# Ontology Modelling for Materials Science Experiments

Mehwish Alam[1,2], Henk Birkholz[3], Danilo Dessì[1,2], Christoph Eberl[4], Heike Fliegl[5], Peter Gumbsch[4], Philipp von Hartrott[4], Lutz Mädler[3], Markus Niebel[4], Harald Sack[1,2], Akhil Thomas[4]

[1] FIZ Karlsruhe – Leibniz Institute for Information Infrastructure, Germany
[2] Karlsruhe Institute of Technology, Institute AIFB, Germany
[3] Leibniz Institute for Materials Engineering IWT, Germany
[4] Fraunhofer IWM, Freiburg, Germany
[5] Karlsruhe Institute of Technology, Institute of Nanotechnology, Germany

**Abstract.** Materials are either enabler or bottleneck for the vast majority of technological innovations. The digitization of materials and processes is mandatory to create live production environments which represent physical entities and their aggregations and thus allow to represent, share, and understand materials changes. However, a common standard formalization for materials knowledge in the form of taxonomies, ontologies, or knowledge graphs has not been achieved yet. This paper sketches major efforts in modelling an ontology to describe materials science experiments. It describes what is expected from the ontology by introducing a use case where a process chain driven by the ontology enables the curation and understanding of experiments.

**Keywords:** Materials Science, Ontology Design, Data Curation

## 1 Introduction

The discipline Materials Science and Engineering (MSE) promises solutions to modern societal challenges, including climate change and resource scarcity. However, the complexity of the lifecycles of materials and their diversity poses several challenges in the management of materials' knowledge for a comprehensive sharing and understanding among various MSE disciplines.

Many experiments are conducted to study materials' behavior, which generates a variety of data, describing manufacturing process settings, material properties, and further MSE parameter. The sharing and interoperability of MSE findings are mainly achieved through the exchange of not standard and often not well-documented files [2]. They are often hardly processable and understandable by humans and machines, thus limiting the potential to support all stakeholders in their tasks. Therefore, modelling MSE data with formal semantics is crucial to consider a variety of MSE facets (e.g., multidisciplinarity or spatial inhomogeneity) to provide a better understanding and support the creation of new materials. A common and shared representation for material knowledge in the

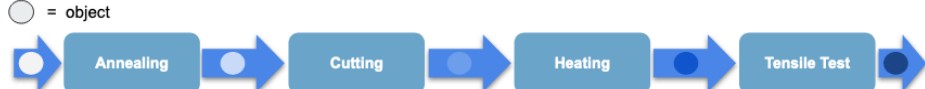
Fig. 1: A process chain with four processes.

form of taxonomies, ontologies, and knowledge graphs has not been achieved yet. Challenges arise in the representation of dynamic events that occur when materials change their state due to manufacturing processes. Existing attempts try to represent top-level knowledge about materials properties and structures [1] with the objective to enable seamless data integration and sharing [4].

Recent ontologies are paving the road for the MSE data interoperability by providing a common ground to describe materials. For example, this challenge is currently being addressed by several communities including the European Materials Modelling Council[6] (EMMC) which develops the European Materials & Modelling Ontology[7] (EMMO) [3] an ontology developed to describe classical and quantum physics. It focuses on high-level properties of materials and manufacturing processes, and extensions to model specific use cases are required. A more recent effort in the MSE domain is given by the Materials Design Ontology (MDO) [2] which has the objective to make different outcomes generated by calculations interoperable. MDO introduces relations between materials' properties and materials' structures, but does not relate their transformations to process parameters. Hence, the description of materials manufacturing might result incomplete.

## 2    Scenario and Vision

Imagine having a process chain. An object undergoes processes that transform the object's material structures, see Figure 1. In this scenario, it is crucial to track how processes are performed and how objects change when describing materials' transformations. In detail, transformative processes (e.g., manufacturing processes) lead to changes in objects' status (i.e., materials' properties and materials' structures) according to their individual process parameters, thus creating new entity objects as an output.

Figure 2 shows a high-level sketch that represents the main top classes and object properties of the ontology under development. Every process comes with its own parameters that represent all the required variables.

Processes that can transform some materials' structures are represented by the class `pmd:ManufacturingProcess`, processes which make analysis are represented by the class `pmd:AnalysisProcess`; however, they can still transform the object (e.g., a Tensile Test process), and therefore, they might also be transformative. Process parameters might have various effects on the materials and, therefore, the relations between processes to materials' structures are required.

---

[6] https://emmc.eu/
[7] https://github.com/emmo-repo/EMMO

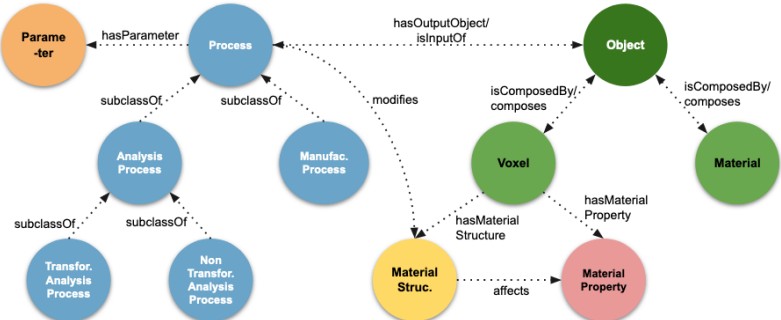

Fig. 2: Visualization of the main elements that constitute the ontology.

This can also be done at different granularity levels depending on the requirements of the application scenario. For example, the problem of locally heterogeneous materials in an object requires separating the object into a certain number of volumetric sub-areas made by the same material (`pmd:Material`), so called voxels (`pmd:Voxel`). Therefore, the voxel becomes the object of the ontology and describes the conditions of the material with its individually experienced process parameters. Object properties are defined to describe how `pmd:Process` modifies `pmd:MaterialStructure` affecting `pmd:MaterialProperty`, thus allowing machine and humans to understand what is performed in the experiment.

## 3 Practical Use of the Ontology

MSE ontologies will enable MSE scientists to curate, describe, share, and optimize experiments. An application example is given in Figure 3. In Figure **??**, there are 2 processes: $c_0$ and $h_0$ instances of `pmd:Cutting` and `pmd:Heating`, respectively. There are 3 `pmd:Object` namely $o_0, o_1, o_2$; $o_1$ is originated from $o_0$ and $o_2$ is originated from $o_1$. $o_0$ has a `pmd:Geometry` $g_0$ and a `pmd:Microstructure` $m_0$. In the example, `pmd:Cutting` represent cutting processes that are transformative for the geometry, `pmd:Heating` processes that are transformative for the microstructure. When $o_0$ undergoes $c_0$, it is transformed in $o_1$; since $c_0$ does not transform the microstructure, it is preserved in $o_1$ (edge $a$). However, $o_1$ will have a different geometry i.e., $g_1$. When $o_1$ undergoes $h_0$, it is transformed in $o_2$. In this case, the geometry is not transformed and, therefore, $o_2$ has the same geometry of $o_1$ i.e., $g_1$ (edge $b$). Thus, the preservation of materials' structures and properties can be defined by means of description logics and Semantic Web Rule Language (SWRL) rules, which helps automatic reasoning on experimental data e.g., to find inconsistencies. For example, a voxel with a different microstructure after a cutting process raises an inconsistency. At the same time, this semantics helps to create connections between voxels involved in a process chain, thus enabling reasoning on the process-object relations. The reader can find a toy example of this practical implication in github[8].

---

[8] `https://github.com/ISE-FIZKarlsruhe/pmd-onto-poster`

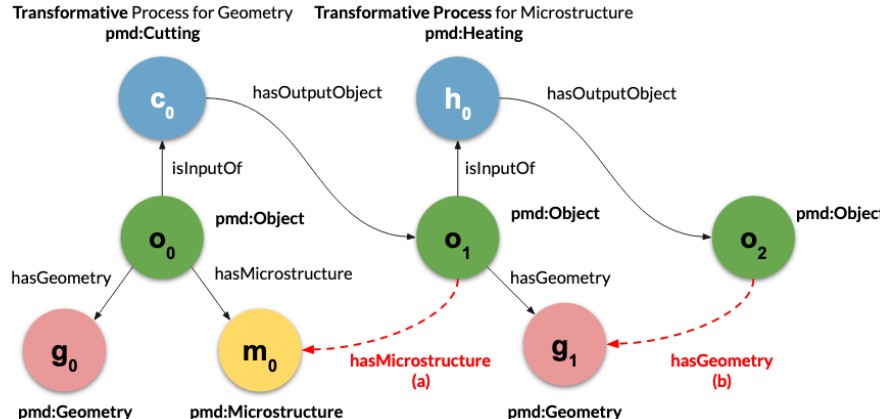

Fig. 3: Materials' structures and properties preservation in a process-chain.

## 4    Conclusions and Outlook

This paper introduces the vision to model process chains and MSE experiments through an ontology with the long term goal of studying materials transformation. Perspectively, the data modelled by specific use case ontologies will enable the curation and preservation of data as well as the possibility to interpret various outcomes. These ontologies are being developed within the Plattform MaterialDigital[9]. They will enable a substantial step towards the provision of findable, accessible, interoperable, and reusable MSE data.

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
