# OpenReview forum: "Ontology Modelling for Materials Science Experiments"
_eswc-conferences.org/ESWC/2021/Conference/Poster_and_Demo_Track — Submitted to ESWC2021 P&D_

### Official Review · AnonReviewer4 · 2021-04-13
**A materials science experiment ontology sounds necessary**

**Rating:** 6
**Confidence:** 3

**Review:**

This paper proposes an ontology for materials science. The structured of the ontology is sketched in enough detail and is available through the GitHub repo included in the paper, which contains plenty of detail. As usual, when the discussion topic is a specific ontology, something that is very important in order to prevent the whole thing from becoming just an intellectual exercise is use cases and actual applications that illustrate the advantages the ontology brings about. This is the part that I miss in this paper, beyond the generic motivation to why this ontology is necessary.

**Anonymity:**

Yes, I would like my review to remain anonymous.

---

### Official Review · AnonReviewer3 · 2021-04-14
**Too early work; not yet interesting**

**Rating:** 4
**Confidence:** 3

**Review:**

This poster submission argues for the need for an ontology describing concepts (primarily processes and materials) in materials science and engineering, and describes an under-development ontology filling this need.

I do not believe this work is suitable for a poster presentation. It presents some early and rough ideas for future work. But it isn't novel or substantial enough of a contribution to trigger significant discussions at the poster session, nor enough of a call-to-action for others to get involved.

Once this work is more mature and developed it might form the basis for a valuable resource and/or in-use track contribution.

**Anonymity:**

Yes, I would like my review to remain anonymous.

---

### Official Review · AnonReviewer2 · 2021-04-14
**Interesting to foster the discussion about an ontology under development**

**Rating:** 6
**Confidence:** 4

**Review:**

The authors present an initial step towards an ontology to model processes and their outcomes in the Material Science domain. While the work seems to be initial steps, it already seems to capture a good part of the required knowledge about the experiments, and it might be interesting in order to foster dicusssions.

From outside the domain, the model of the processes seems to be completely sequential. Could it be a limitation of the model in order to capture different types of experiments performed with the materials? Maybe the adoption of a BPM-oriented model for this could enhance the semantics of the sequences of what actually happened in the experiments (e.g., to model problems such as half of the material didn't reach the required temperature and the outcome was X, but the other half did so and the outcome was Y).

Typo:
page 3 => Figure ??

**Anonymity:**

Yes, I would like my review to remain anonymous.

---

### Official Review · AnonReviewer1 · 2021-04-14
**Not mature enough but promising work on modelling an ontology for Materials Science experiments**

**Rating:** 4
**Confidence:** 4

**Review:**

This paper aims at introducing an ontology to describe materials science experiments and more specifically chains of processes to analyze and/or transform objects composed of some materials.
The paper is well written and the topic is of interest for the ESWC community. However the work does not seem yet mature enough to be presented:
-	the method for designing the targeted ontology is not discussed and the involvement of Materials Science experts is unclear,
-	 the relationship (alignment)of the presented ontology  with the EMMO and MDO ontologies (and other possible candidate ontologies to describe processes or measurements) is not discussed; the precise scope of the ontology is therefore unclear,
-	the expressive power of the presented ontology is unclear: the example graph in fig 3 shows how changes in the structure or geometry of an object are handled but is too simple (the reader is expecting an RDF graph with specific characteristics associated to the 2 processes and 3 objects); no class definition or SWRL rule and no SPARQL query exploiting the data are presented.
-	an OWL ontology is publicly available on github but comprises only 6 classes and 6 properties, I do not find some classes presented in Fig 2. And there are only 2 SWRL (which are not presented in the paper).


**Anonymity:**

Yes, I would like my review to remain anonymous.

---

### Official Review · Program_Chairs · 2021-04-18
**Metareview: Reject (Promising work, but too preliminary)**

**Rating:** 4
**Confidence:** 5

**Review:**

This was a borderline paper with two reviewers arguing for reject, and two reviewers leaning towards accept, with the average scores overall leaning towards reject. While the reviewers agree that the proposal of an ontology for material sciences is an interesting one, there are concerns that the work is still too preliminary, and too many aspects of the proposed ontology remain unclear, in terms of its creation, its expressivity, etc. Taking into consideration all of the comments, while we think that this is promising work, and while the P&D session historically welcomes discussion of promising ongoing work, based on the reviewers' comments we think that it is perhaps a little too early for the current paper to be accepted. We think that this may make an excellent contribution to a future venue when the work has further matured.

**Anonymity:**

Yes, I would like my review to remain anonymous.

---

### Decision · Program_Chairs · 2021-04-19

Reject